# Treatment, Prognostic Markers, and Survival in Thymic Neuroendocrine Tumors, with Special Reference to Temozolomide-Based Chemotherapy

**DOI:** 10.3390/cancers16142502

**Published:** 2024-07-10

**Authors:** Zixuan Cheng, Fuhuan Yu, Ruao Chen, Lingjun Cui, Yingying Chen, Chao Deng, Yanfen Shi, Huangying Tan

**Affiliations:** 1Graduate School, Beijing University of Chinese Medicine, Beijing 100029, China; 20210941348@bucm.edu.cn (Z.C.);; 2Department of Integrative Oncology, China-Japan Friendship Hospital, Beijing 100029, China; 3Department of Pathology, China-Japan Friendship Hospital, Beijing 100029, China

**Keywords:** thymic neuroendocrine tumors, prognostic markers, temozolomide-based chemotherapy, survival

## Abstract

**Simple Summary:**

Thymic neuroendocrine tumors (Th-NETs) are rare, with limited research on prognostic markers and treatments. Our retrospective study evaluated clinicopathological characteristics and therapeutic strategies for Th-NETs, filling gaps in existing studies. We highlighted the importance of biomarkers such as serum neuron-specific enolase, inflammatory factors, and oxygen 6-methylguanine-DNA methyltransferase, as well as the roles of surgical interventions, postoperative therapies, and temozolomide-based chemotherapy in improving patient survival. Our findings confirmed the crucial role of surgical treatment and the potential benefits of postoperative adjuvant radiotherapy. Furthermore, temozolomide-based chemotherapy demonstrated efficacy in Th-NETs treatment, with the neutrophil-to-lymphocyte ratio and serum neuron-specific enolase proving to be valuable prognostic markers necessitating further research.

**Abstract:**

Background: Thymic neuroendocrine tumors (Th-NETs) are rare and aggressive, with a scarcity of research on predicting patient prognosis. Our study aimed to assess the impact of prognostic markers and temozolomide (TMZ)-based chemotherapy on survival in Th-NETs. Methods: We retrospectively reviewed the medical records of patients diagnosed with Th-NETs between 2013 and 2023 at our institution. We collected clinicopathological data, including tumor pathological grading, staging, serum concentrations of neuron-specific enolase (NSE) and pro-gastrin-releasing peptide, levels of inflammatory factors, and expression of oxygen 6-methylguanine-DNA methyltransferase (MGMT). Treatment details (such as surgery and chemotherapy) and survival outcomes were also documented. Results: A total of 32 patients were included in our study after excluding those without complete available information. The median progression-free survival (PFS) was 12.5 months (95%CI, 8–16 months) for 19 patients who received TMZ-based chemotherapy. Twenty-one patients underwent surgery as the primary treatment, demonstrating a median disease-free survival (DFS) of 51.0 months. The inflammatory factor neutrophil-to-lymphocyte ratio (NLR) was an independent prognostic indicator of DFS in postoperative patients and PFS in TMZ-treated patients. The overall 3-, 5-, and 10-year survival rates were 86.6%, 69.5%, and 33.8%, respectively. Ki67 level exceeding 10% (*p* = 0.048) and absence of surgical resection (*p* = 0.003) were significantly associated with worse overall survival (OS). Conclusion: Surgical treatment was currently the primary method for treating Th-NETs, and postoperative adjuvant therapy was an essential consideration for specific patient cohorts. Despite widespread positive MGMT expression, TMZ-based chemotherapy showed promise. Some potential prognostic biomarkers such as NLR and NSE need more attention.

## 1. Introduction

Thymic neuroendocrine neoplasms (Th-NENs) are rare and highly heterogeneous malignant tumors originating from neuroendocrine cells, with an estimated incidence of 0.02 per 100,000 individuals. Th-NENs, along with thymoma and thymic carcinoma, all fall under the category of thymic epithelial tumors (TETs) [1]. Th-NENs represent a minority of thymic tumors (2–5%) and an even smaller fraction of all neuroendocrine neoplasms (<0.5%) [2]. They are highly aggressive and have a poor prognosis, with around 50% of patients failing to reach a 5-year survival rate [3,4].

Early symptoms of Th-NENs are not prominent, with most patients presenting with cough, chest pain, and superior vena cava syndrome due to the rapid progression of mediastinal tumors [5]. Functional Th-NENs may be associated with ectopic ACTH syndrome, while carcinoid syndrome is relatively rare. According to the 2022 WHO pathological classification, Th-NENs can be divided into well-differentiated neuroendocrine tumors (NETs) and poorly differentiated neuroendocrine carcinomas (NECs). NETs are further categorized into typical carcinoids (TCs), atypical carcinoids (Acs), and carcinoids/NETs with elevated mitotic counts and/or a high Ki67 proliferation index [6]. Currently, surgery is the preferred primary treatment for all potentially radically resectable, well-differentiated thymic neuroendocrine tumors (Th-NETs). However, even after radical resection, there is still a noteworthy risk of local recurrence and distant metastasis [5,7]. Moreover, the elusive characteristic of the disease often leads to a late diagnosis at an advanced stage when curative surgery is unfeasible. Nevertheless, the lack of agreement on treatment decision making in the advanced stages and the benefit of postoperative adjuvant therapy is attributable to the sporadic incidence and limited clinical research data of Th-NENs. Clinical management for Th-NENs is often informed by experiences gained with lung neuroendocrine tumors (L-NETs) [8]. While guidelines recommend temozolomide (TMZ)-based chemotherapy for Th-NENs, there is a lack of specific recommendations based on solid clinical research regarding its practical implementation. Furthermore, there is a need for further investigation to examine the association between the prognosis of Th-NENs and clinicopathological characteristics such as tumor stage, histological grading, and Ki67, which show close associations with prognosis in NETs from other sites [4].

In this context, we retrospectively analyzed the clinical and pathological characteristics of patients with Th-NETs treated at our institution over the past decade, aiming to provide valuable insights into the clinicopathological features, prognostic factors, and therapeutic strategies for Th-NETs.

## 2. Materials and Methods

The retrospective study was approved by the clinical research ethics committee of the China–Japan Friendship Hospital (approval 2023-KY-096). Patients’ data were obtained from the medical database of the China–Japan Friendship Hospital and analyzed anonymously and exclusively for academic research purposes.

### 2.1. Patients

We conducted a retrospective analysis of clinical and pathological data from patients with well-differentiated Th-NETs who were treated at the China-Japan Friendship Hospital between July 2013 and June 2023. Patients lacking complete available information (demographic, clinicopathologic characteristics, and follow-up) were excluded from this study. Both biopsy and resection specimens were centrally reviewed by two expert pathologists in neuroendocrine tumor pathology. Treatment strategies were meticulously personalized, incorporating surgery, radiotherapy, and pharmacotherapy, based on detailed evaluations conducted by a multidisciplinary team. This team consisted of a chief internal medicine physician with a specialization in neuroendocrine neoplasms, a thoracic surgeon, two pathologists, and a radiation oncologist, facilitating a comprehensive and expert-led approach to patient management.

### 2.2. Follow-up and Outcomes

Demographic and clinicopathological characteristics at baseline were retrieved from the medical and pathology record system of the China–Japan Friendship Hospital including age, gender, tumor size, tumor grade, tumor stage, hematological and biochemical parameters, immunohistochemical information, and treatment. The final follow-up visits were conducted on 30 November 2023. The dates of recurrence and metastasis and any survival information were obtained from medical records and follow-up calls. Tumor response was evaluated using CT scans and/or MRI based on Response Evaluation Criteria in Solid Tumors (RECIST) 1.1 [9], including complete remission (CR), partial remission (PR), stable disease (SD), and progressive disease (PD). Overall survival (OS) was defined as the duration from the diagnosis to death or last follow-up for surviving patients, while disease-free survival (DFS) referred to the period from surgery to the occurrence of the first instance of tumor recurrence (local and distant metastases); progression-free survival (PFS) was defined as the duration from the initial treatment to the identification of disease progression or death or the end of follow-up.

### 2.3. Clinicopathological Characteristics

According to the 2022 WHO Classification of Endocrine and Neuroendocrine Tumors [6], Th-NETs were classified into typical carcinoid (TC), atypical carcinoid (AC), and carcinoids/NETs with elevated mitotic counts and/or Ki67 proliferation index. The phrase “carcinoids/NETs with elevated mitotic counts and/or Ki67 proliferation” referred to an updated classification in the 2022 WHO classification. Its diagnostic criteria included atypical carcinoid morphology, a higher mitotic count (>10 mitoses per 2 mm^2^), and/or a higher Ki67 index (>30%). Tumor stages were reassessed based on the American Joint Committee on Cancer TNM (7th edition) [10] and the Masaoka systems [11]. All pathological grades and tumor stages were reassessed by senior physicians according to the aforementioned standards. The tumor size was determined by measuring the pathological diameter in patients who received surgical intervention and by assessing chest computed tomography (CT) scans in patients who did not receive surgical intervention. The absolute lymphocyte count (ALC) is defined as the total number of lymphocytes in the blood. The neutrophil-to-lymphocyte ratio (NLR) is calculated as the ratio of neutrophils to lymphocytes. The platelet-to-lymphocyte ratio (PLR) involves the ratio of platelets to lymphocytes. Finally, the lymphocyte-to-monocyte ratio (LMR) is determined by dividing the number of lymphocytes by the number of monocytes. The optimal cut-off values were determined by analyzing the ROC curves of ALC, NLR, PLR, and LMR, following which patients were stratified into groups accordingly. All patients with Th-NETs in this study underwent systematic evaluations and received clinical diagnoses in accordance with the Clinical Practice Guidelines for Multiple Endocrine Neoplasia Type 1 (MEN1) [12]. For patients presenting with a family history or two or more MEN1-associated endocrine tumors, such as parathyroid adenomas, gastroenteropancreatic neuroendocrine tumors, and pituitary tumors, genetic testing using the next-generation sequencing (NGS) method was conducted on blood samples. The detection of a MEN1 germline mutation confirmed the diagnosis of MEN1 syndrome. Immunohistochemical analysis was employed to evaluate the levels of somatostatin receptor 2 (SSTR2) and oxygen 6-methylguanine-DNA methyltransferase (MGMT) expression.

### 2.4. Statistical Analysis

All statistical analyses were conducted using SPSS version 26.0 (IBM, Armonk, NY, USA) and R statistical software (R, version 3.5.1; R Foundation for Statistical Computing, Institute for Statistics and Mathematics, Vienna, Austria). Survival curves were calculated by the Kaplan–Meier method, and univariate analyses were conducted using the log-rank test. Prognostic factors were evaluated by examining clinicopathological variables upon univariate analysis. Variables with a *p*-value < 0.05 upon univariate analysis were included in multivariate analysis using Cox proportional hazard regression models. A *p*-value less than 0.05 was considered statistically significant.

## 3. Results

### 3.1. Clinicopathological Characteristics

A total of 74 patients diagnosed with well-differentiated Th-NETs attended our institution and were documented in the medical record system, but only 32 of them had complete available information (demographic, clinicopathologic feature, and follow-up) and were thus included in this study. The patients consisted of 24 (75.0%) male and 8 (25.0%) female patients, with a mean age of 47.09 ± 10.21 years (range, 22–66 years). The most prevalent initial symptom was chest discomfort, reported by 11 (34.3%) patients, followed by superior vena cava syndrome in 4 (12.5%), and swollen lymph nodes in 3 (9.4%) patients. Ten (31.3%) patients’ conditions were detected coincidentally during routine physical examinations. Serum levels of neuron-specific enolase (NSE) ≥ 16.3 ng/mL were observed in 15 (46.9%) patients, and serum pro-gastrin-releasing peptide (Pro-GRP) levels ≥ 67.42 ng/mL were observed in 16 (50.0%) patients. Among the tumors analyzed, 3 (9.4%) were TCs, 25 (78.2%) were ACs, and 4 (12.5%) were carcinoids/NETs with elevated mitotic counts and/or Ki67 proliferation index. The average tumor size was 6.8 ± 0.5 cm (range 2.0 cm–15.0 cm). A total of 6 (18.8%) patients had MEN1 syndrome. Additionally, 12 (37.5%) patients presented positive expression of SSTR2, while 27 (84.4%) presented positive expression of MGMT. Distant metastases were detected in 10 patients (31.3%) upon diagnosis. The demographic and clinicopathologic characteristics of the patients are shown in Table 1 (Appendix A provides detailed data).

### 3.2. Treatments

A total of 21 patients underwent surgical intervention, with 10 of them not receiving postoperative treatment. Among the patients who received postoperative treatment, seven were given chemotherapy alone, one received radiotherapy alone, and three received a combination of chemotherapy and radiotherapy. Among those who received only postoperative chemotherapy, four patients were treated with the EP regimen (etoposide and cisplatin), two patients with the EC regimen (etoposide and carboplatin), and one patient with the CapTem regimen (capecitabine and temozolomide). Of the three patients who received combined postoperative chemotherapy and radiotherapy, one received the EP regimen, one the GC regimen (gemcitabine and cisplatin), and one a regimen including paclitaxel and cisplatin. Notably, only one patient, who underwent surgical resection of the primary mass, showed distant metastases upon diagnosis, with the metastatic site identified as a solid pulmonary nodule. Of the 11 patients who did not receive surgical intervention, nine presented with distant metastases upon diagnosis. Seven patients received chemotherapy as first-line treatment (four with the EP regimen, three with the CapTem regimen), two patients received chemotherapy (CapTem regimen) combined with radiotherapy, and two patients received somatostatin analogs (SSAs) and everolimus, respectively. Upon tumor recurrence or progression, patients commonly received TMZ-based chemotherapy (n = 8), Surufatinib (n = 7), SSA (n = 4), and platinum-based chemotherapy (n = 3). Among the advanced-stage patients, 19 underwent TMZ-based chemotherapy (CapTem regimen), with 5 receiving it as first-line treatment, 8 as a second-line treatment, and 6 as third-line or subsequent treatment. Out of these patients who received TMZ-based chemotherapy, nine individuals (47.4%) had previously undergone surgery. Initial efficacy assessments revealed three patients achieving PR, while two patients experienced PD. The treatment strategies of 32 patients are shown in Table 2 and Appendix A, and the responses to TMZ-based chemotherapy are shown in Table 3.

### 3.3. Survival

All 32 patients had complete follow-up data, with a median follow-up of 48.4 months (range: 11.2–123.7 months), during which 11 deaths occurred. The median OS was 123.7 months (range: 85.2–162.2 months). The overall rates at 3, 5, and 10 years were 86.6%, 69.5%, and 33.8%, respectively. Among the 21 patients who underwent surgery, 15 experienced recurrence and/or distant metastasis, with bone metastasis being the most common (n = 6). Median DFS was 51 months (range: 15.2–86.8 months), with corresponding rates of 56.3% at 3 years and 30.4% at 5 years. All nonsurgical patients experienced disease progression after first-line treatment. The median PFS for all patients was 17.4 months (range: 2.3–34.9 months), with corresponding rates of 40.1% and 23.6% at 3 and 5 years. Among those who received TMZ-based chemotherapy, the median PFS (TMZ-PFS) was 12.5 months (range: 3.6–21.4 months), and the median OS (TMZ-OS) was 62.4 months (range: 38.2–86.6 months). Survival curves are illustrated in Figure 1a–e.

### 3.4. Prognostic Factors

Univariate analysis unveiled several factors strongly associated with prolonged OS, including low Ki67 (≤10%; *p* = 0.004; Figure 2a), low NSE level (<16.3 ng/mL; *p* = 0.025), absence of distant metastasis at diagnosis (*p* = 0.003), early TNM staging (stage I–II; *p* = 0.044), early Masaoka staging (stage I–II; *p* = 0.044), and surgical intervention (*p* = 0.002; Figure 2b). Conversely, gender, mitotic index, pathological grading, tumor size, Pro-GRP levels, and inflammatory factor levels (ALC, NLR, LMR, and PLR) displayed no significant associations with OS. Multivariate analysis revealed that Ki67 > 10% (HR 9.54; 95%CI 1.55–58.58; *p* = 0.007) and absence of surgical resection (HR 0.24; 95%CI 0.02–3.92; *p* = 0.003) were independent risk factors for poorer OS (Table 4).

Similarly, univariable analysis of DFS indicated a significant association between prolonged DFS and several factors including TNM staging stage I–II (*p* = 0.049; Figure 3a), Masaoka staging stage I–II (*p* = 0.049; Figure 3b), adjuvant postoperative therapy (*p* = 0.027; Figure 3c) and lower NLR levels (<1.94; *p* = 0.020; Figure 3d). Multivariable analysis further corroborated the importance of early TNM staging, and early Masaoka staging, low NLR levels, and adjuvant postoperative therapy emerged as pivotal factors for bolstering DFS postoperatively (Table 5).

Univariable analysis indicated that the distant metastasis at diagnosis (*p* < 0.001; Figure 2c), TNM staging (*p* = 0.001), Masaoka staging (*p* = 0.001), and surgical intervention (*p* < 0.001) correlated with PFS of all Th-NET patients. However, in the multivariate analysis, only the distant metastasis at diagnosis (*p* = 0.001) was an independent prognostic factor.

In patients receiving TMZ-based chemotherapy, male gender (*p* = 0.028; Figure 4a), lower NLR (<1.94; *p* = 0.023; Figure 4b), and lower PLR (<106.1; *p* = 0.044) were associated with prolonged PFS (Table 6). Furthermore, gender and inflammatory factor NLR levels were associated with PFS in patients treated with TMZ-based chemotherapy in the multivariate analysis. Additionally, Ki67 (*p* = 0.031), distant metastasis at diagnosis (*p* = 0.049), surgical intervention (*p* = 0.010), and PLR level (*p* = 0.048) influenced OS with TMZ-based chemotherapy in univariable analysis, although this result was not statistically significant upon multivariate analyses.

## 4. Discussion

Th-NETs are a rare subset, accounting for less than 0.5% of all neuroendocrine tumors [8]. The limited occurrences of Th-NETs have hindered clinical research due to small sample sizes, inconsistent clinical evaluations, and reliance on retrospective analysis of medical records. A more robust dataset is necessary to gain a thorough understanding of the clinicopathological characteristics, prognostic markers, and treatment approaches for Th-NETs. Our investigation revealed a male predominance in Th-NET cases, typically presenting around the age of 47.09 years. The primarily reported symptom was chest discomfort, with common metastatic sites including the lungs, bones, mediastinum, and cervical lymph nodes. The gender distribution and age profile of our patient cohort closely mirrored those delineated in the ESMO guidelines [8]. Additionally, the primary symptoms reported and the preferred sites of metastasis aligned with results from previous investigations [7,13,14].

In thoracic neuroendocrine tumor studies, serum chromogranin A (CgA) has been predominantly discussed in the context of L-NETs, albeit with limited sensitivity and prediction accuracy [15]. Elevated serum tumor markers NSE and Pro-GRP are commonly associated with poorly differentiated neuroendocrine carcinomas (NECs) [16,17]. However, only one small-sample study reported elevated NSE levels in approximately 35.7% of Th-NETs [18]. In our study, elevated serum NSE levels were observed in 53.1% of patients, while elevated Pro-GRP levels were noted in 50.0% of patients. Univariate analysis suggested a possible association between elevated NSE levels and worse OS. Serum NSE and Pro-GRP may be potential biomarkers in the clinical application of well-differentiated Th-NETs. Additionally, our study examined the levels of inflammatory factors, including ALC, NLR, LMR, and PLR, in Th-NETs, which were scarcely discussed in existing studies. Notably, the prognostic value of NLR has been observed in thymic epithelial tumors [19].

Studies have indicated that approximately 4.76% to 25% of Th-NETs are associated with MEN1 syndrome, often presenting as non-functional tumors and more frequently observed in males [20,21,22]. In our study, we found that 18.8% of Th-NETs patients had MEN-1 syndrome, with only two of them being female. Given the absence of a standardized staging system for Th-NETs, we adhered to the ESMO guidelines by utilizing both TNM stage and Masaoka stage classifications for staging purposes [8]. Our analysis showed that the two staging systems were basically the same in terms of their impact on prognosis. SSTR2 and MGMT are well-recognized predictive markers for SSA and temozolomide efficacy, respectively, and are commonly discussed in gastroenteropancreatic and pulmonary neuroendocrine tumors [23,24,25,26]. However, their expression in Th-NETs has not been previously reported. In this study, immunohistochemical methods were applied to access the expression levels of SSTR2 and MGMT in Th-NETs. The findings revealed that SSTR2 had a positivity rate of 37.5%, whereas MGMT showed a significantly higher positivity rate of 84.4%.

In this study, the survival rates at 3, 5, and 10 years for patients with Th-NETs were 86.6%, 69.5%, and 33.8%, respectively, consistent with the findings reported by Filosso et al. [7] and Fang et al. [13]. Despite these similarities, establishing a consensus on the prognostic factors impacting OS across various studies remains a significant challenge. Sullivan et al. [27] assessed the SEER database and identified surgical resection, tumor size, and Masaoka staging as independent prognostic factors influencing OS. Meanwhile, in the study conducted by Fang et al. [13], the only independent predictors linked to OS were surgical resection and the completeness of resection (R0 vs. R1-R2). The results from the study by Filosso et al. [7] corroborate previous research, identifying surgical resection, completeness of resection, TNM staging, and pathological grade as independent prognostic factors for OS. In our study, the status of surgical resection (*p* = 0.003) was also an independent prognostic factor for OS, reinforcing surgical resection as the optimal treatment strategy for Th-NETs, as corroborated by multiple clinical investigations. Additionally, our study indicated that a Ki67 index larger than 10% (*p* = 0.007) was an independent risk factor for poorer OS. However, the Ki67 proliferation index was served as an adjunct to pathological grading, and its role as a prognostic marker in lung NETs was still controversial [28,29], with even fewer reports in Th-NETs [4]. If it is to be employed as a supplementary method to pathological grading for predicting clinical outcomes, further research support is essential. We further explored the correlation between tumor size and OS using 7 cm as a cutoff, aligning with methodologies employed in several large retrospective studies [14,27,30]. Unfortunately, our analysis did not reveal a significant correlation, potentially due to the limited size of our sample.

Despite aggressive surgical management of patients, rates of local recurrence and/or distant metastases remain high, which is one of the main reasons for the poor long-term outcomes observed in Th-NET patients. A concerning 71.42% of patients in our study who underwent surgical removal of the tumor subsequently experienced local recurrences and/or distant metastases, with a median DFS of 51.0 months. Multivariate analysis revealed that TNM staging, Masaoka staging, postoperative adjuvant therapy, and NLR levels were independent prognostic factors for DFS. Notably, there was a scarcity of clinical research on Th-NETs that investigated DFS. The latest investigation into DFS, conducted by Zhai et al. [31], identifies Masaoka staging and the presence of major vascular invasion in postoperative pathology as critical factors that significantly impact the rates of postoperative local recurrence and metastasis. Debate persists regarding the impact of postoperative adjuvant therapy on prognosis. Research by Tiffet et al. [32] suggested that patients receiving postoperative radiotherapy following complete tumor resection exhibit improved outcomes (no recurrence), but more studies have shown that postoperative adjuvant chemotherapy/radiotherapy offers no benefit to Th-NETs patients [7,27,31]. Our results demonstrate that postoperative adjuvant therapy can enhance DFS, possibly due to the higher proportion of patients receiving adjuvant chemotherapy and the presence of stage III individuals among those who underwent surgical resection. The predictive utility of the NLR for DFS has not yet been documented in Th-NETs, although it has been previously discussed in thymic epithelial tumors [19]. Moreover, the correlation between the NLR and prognosis in medullary thyroid carcinoma (MTC), another rare neuroendocrine tumor, has been extensively discussed. Roberta Modica et al. observed significant changes in NLR levels between the preoperative and postoperative periods, which may be linked to the tumor’s inflammatory response [33]. Multivariate logistic regression analysis in the study of Xu et al. demonstrated that NLR was an independent predictor of metastasis [34]. Unfortunately, no one has confirmed that NLR can be used as an independent predictor of DFS in the field of MTC. Therefore, although our multivariable analysis results confirmed the significant correlation between NLR and DFS in Th-NETs, the clinical application of NLR in neuroendocrine tumors still needs further exploration.

Beyond evaluating OS and DFS, this study also specifically examined the prognosis of patients treated with TMZ. TMZ-based chemotherapy is emerging as a promising therapeutic option for Th-NETs with high proliferative rates, despite the limited research on the topic. While current guidelines endorse SSA as the first-line therapy for TC and everolimus for AC, TMZ-based regimens have also shown therapeutic efficacy in both pulmonary and thymic NETs. Reports indicate that approximately 10% to 30% of patients achieve PR, with a median PFS ranging from 5 to 13 months [8]. One of the higher-quality pieces of clinical evidence comes from a multicenter, open-label, single-arm, phase II trial that included patients with large-cell lung neuroendocrine carcinoma and utilized temozolomide as a second-line treatment. The study reported a median PFS of 5.86 months, indicating promising activity [35]. While the majority of research has centered on L-NETs, there were limited studies on Th-NETs. Perry et al. [36] reported the therapeutic outcomes in three cases, and Crona et al. [22] found a median PFS of 20.5 months in eight patients treated with TMZ in their retrospective analysis. Furthermore, in 2019, Wang et al. from our institution reported a median PFS of 8 months in nine patients with thymic ACs treated with the CapTem regimen [37]. Our study analyzed 19 patients who underwent therapies with TMZ, among which 5 received TMZ-based chemotherapy as their first-line treatment. The median PFS for temozolomide-treated patients was 12.5 months, with an overall response rate (ORR) of 15.8%. Reviewing the efficacy of SSAs and everolimus, the first-line treatments recommended by current guidelines, a retrospective study showed that the median PFS for SSA treatment of L-NETs was 11 months [38]. Meanwhile, a prospective study (LUNA trial), which included patients with advanced Th-NETs, reported a median PFS of 8.51 months for SSAs and 12.5 months for everolimus treatment, respectively [39]. Despite its retrospective nature, our study contributes significantly to the understanding of Th-NETs, which are typically marked by poor prognosis and scant treatment options. The demonstrated median PFS of 12.5 months indicates that temozolomide-based chemotherapy is an effective therapeutic option for Th-NETs and warrants further prospective validation. Moreover, MGMT, as a potential predictive biomarker for temozolomide’s effectiveness, has been a focal point of our analysis. Prospective studies in gastrointestinal, pancreatic, and pulmonary NETs have previously demonstrated that negative MGMT expression is associated with a favorable response to temozolomide [40,41]. The expression of MGMT in neuroendocrine tumors (NETs) varies significantly across different anatomical sites. Initially, Kulke et al. [42] in 2009 reported that 51% of pancreatic NETs exhibited a loss of MGMT expression, whereas all pulmonary carcinoid tumor patients (n = 40) demonstrated positive MGMT expression. Further investigations have found that approximately 50–70% of pancreatic NETs show negative MGMT expression. In contrast, the prevalence of MGMT-negative cases in pulmonary and gastrointestinal NETs ranges from 0–15% [24,40,41,43,44]. In Th-NETs, our findings paralleled those observed in L-NETs, with a significant majority of patients (84.4%) exhibiting positive MGMT expression. The limited number of patients with negative MGMT expression prevented a statistical analysis of the correlation between MGMT status and the efficacy of temozolomide. Nonetheless, it is noteworthy that both patients (two out of three) who achieved PR showed negative MGMT expression. Although the majority of Th-NETs patients exhibit positive MGMT expression, our observations still indicate a favorable therapeutic activity of temozolomide-based chemotherapy. Additionally, our study confirms that gender and the level of the inflammatory marker NLR are independent prognostic factors for PFS of patients with TMZ treatment. These insights are crucial for refining the clinical application of temozolomide in Th-NETs management.

Compared to previous studies on the clinicopathological characteristics and prognosis of Th-NETs, our study provides a thorough investigation with extended, continuous follow-up of Th-NETs patients treated at our institution, with detailed compilation of treatment results. We also examined changes in serum tumor markers and inflammatory factors in Th-NET patients, presenting the expression status of SSTR2 and MGMT in pathological evaluations, thereby filling a significant data void in Th-NET research. Furthermore, our focus on the efficacy and prognostic relevance of TMZ-based chemotherapy supports its continued application in the Th-NETs. Nevertheless, the study is constrained by limitations such as a small sample size, the singular nature of the treatment center, and a lack of attention to treatment safety.

## 5. Conclusions

In conclusion, surgical resection remains the primary therapeutic strategy for Th-NETs, and postoperative adjuvant therapy is an essential consideration for specific patient cohorts. Despite the prevalent positive expression of MGMT in Th-NETs, TMZ-based chemotherapy continues to show promising therapeutic potential. Furthermore, the inflammatory marker NLR has shown prognostic significance in both surgically resected and TMZ-treated patients, suggesting its potential as a reliable indicator for predicting treatment response. Serum levels of NSE and Pro-GRP may serve as valuable clinical biomarkers for well-differentiated Th-NETs. However, these conclusions necessitate further substantiation through large-scale clinical studies.

## Figures and Tables

**Figure 1 cancers-16-02502-f001:**
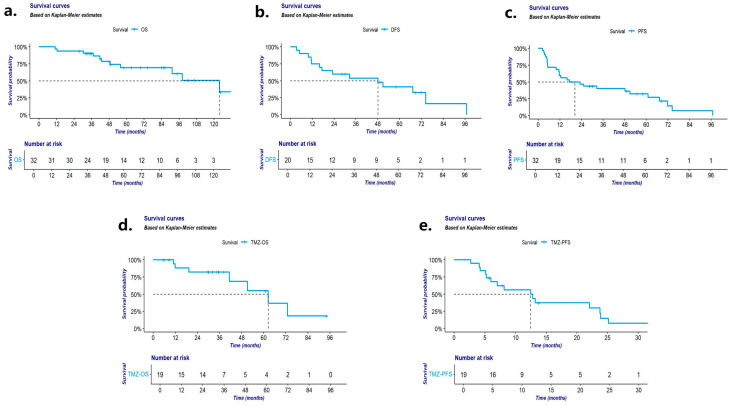
(**a**) OS: overall survival of patients with Th-NETs; (**b**) DFS: disease-free survival of Th-NET patients with surgery approach; (**c**) PFS: progression-free survival of patients with Th-NETs; (**d**) TMZ-OS: overall survival of Th-NET patients with TMZ-based chemotherapy; (**e**) TMZ-PFS: progression-free survival of Th-NET patients with TMZ-based chemotherapy.

**Figure 2 cancers-16-02502-f002:**
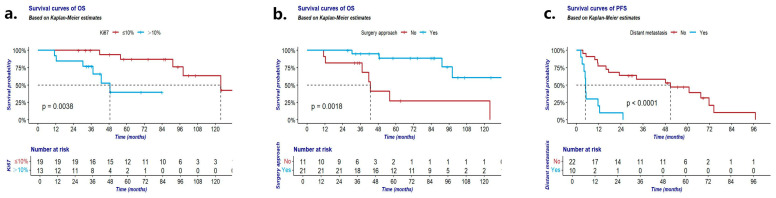
(**a**) OS according to Ki67; (**b**) OS according to surgery approach; (**c**) PFS according to distant metastasis. OS: overall survival of patients with Th-NETs; PFS: progression-free survival of patients with Th-NETs.

**Figure 3 cancers-16-02502-f003:**
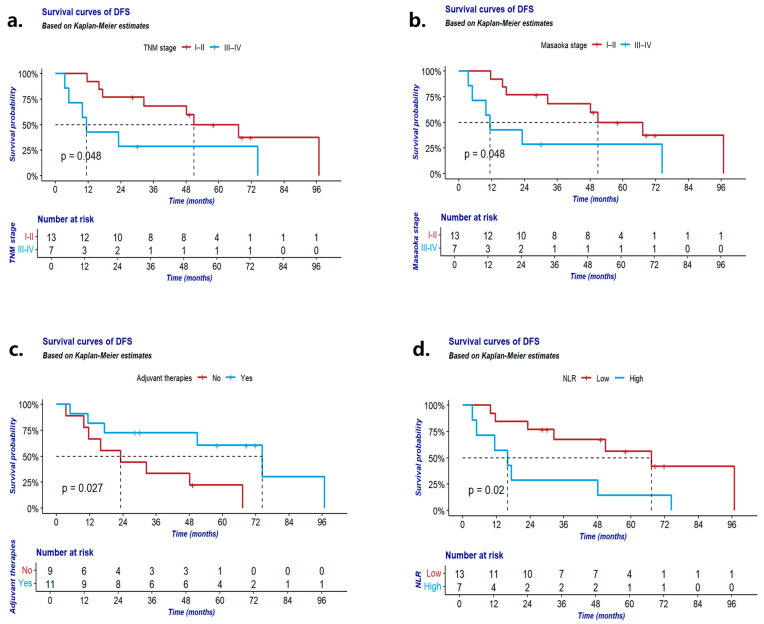
(**a**) DFS according to TNM stage; (**b**) DFS according to Masaoka stage; (**c**) DFS according to adjuvant therapies; (**d**) DFS according to NLR. DFS: disease-free survival of Th-NET patients with surgical approach.

**Figure 4 cancers-16-02502-f004:**
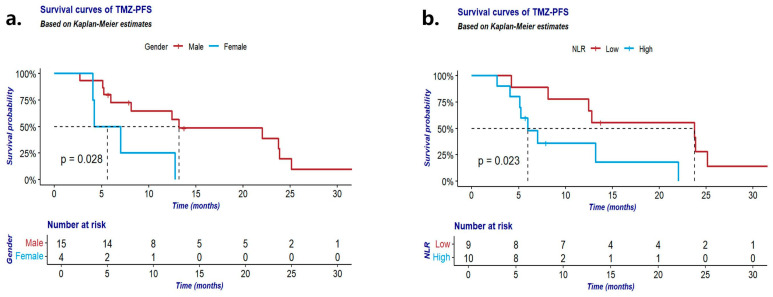
(**a**) TMZ-PFS according to gender; (**b**) TMZ-PFS according to NLR. TMZ-PFS: progression-free survival of Th-NET patients with temozolomide-based chemotherapy.

**Table 1 cancers-16-02502-t001:** Demographic and clinicopathological characteristics of 32 patients with Th-NETs.

Clinicopathological Characteristics		Number of Patients (*n* = 32)	%
Gender	Male	24	75.0
	Female	8	25.0
Age (years)	Mean ± SD, range	47.09 ± 10.21 (22.0–66.0)	
First symptoms	Chest discomfort Superior vena cava syndrome Fever Gastrointestinal symptoms Swollen lymph nodes Ectopic ACTH syndrome None ^a^	11 4 2 1 3 1 10	34.3 12.5 6.3 3.1 9.4 3.1 31.3
NSE (ng/mL) ^b^	<16.3	15	46.9
	≥16.3	17	53.1
Pro-GRP (ng/mL) ^c^	<67.42	16	50.0
	≥67.42	16	50.0
ALC	Mean ± SD, range	1.6 ± 0.5 (0.5–2.5)	
NLR	Mean ± SD, range	2.2 ± 1.1 (0.9–5.8)	
LMR	Mean ± SD, range	4.1 ± 1.8 (1.7–9.8)	
PLR	Mean ± SD, range	137.0 ± 56.3 (30.9–295.9)	
Histological diagnosis	Typical carcinoid	3	9.4
	Atypical carcinoid	25	78.1
	Carcinoids/NETs with elevated mitotic counts and/or Ki67 proliferation index	4	12.5
Mitotic index (/10HPF)	<2	5	15.6
	2–10	23	71.9
	>10	4	12.5
Ki67 (%)	<5.0	1	3.1
	5.0–30.0	28	87.5
	>30.0	3	9.4
Tumor size (cm)	Mean ± SD, range	6.8 ± 0.5 (2.0–15.0)	
Distant metastasis	Yes	10	31.3
	No	22	68.8
TNM stage	I IIIa IIIb IVa IVb	14 3 3 2 10	43.8 9.4 9.4 6.3 31.3
Masaoka stage	I IIa IIb III IVb	7 2 5 7 11	21.9 6.3 15.6 21.9 34.4
MEN-1 syndrome	Yes	6	18.8
	No	26	81.2
SSTR2	Positive	12	37.5
	Negative	20	62.5
MGMT	Positive	27	84.4
	Negative	5	15.6

^a^ Patients without symptoms had lesions detected during routine physical examinations. ^b^ The cut-off value for NSE was selected based on the normal range of serum NSE concentration as reported in the laboratory tests issued by the Department of Laboratory Medicine at our center. ^c^ The cut-off value for Pro-GRP was selected based on the normal range of serum Pro-GRP concentration as reported in the laboratory tests issued by the Department of Laboratory Medicine at our center. NSE: neuron-specific enolase; Pro-GRP: pro-gastrin-releasing peptide; ALC: absolute lymphocyte count; NLR: neutrophil-to-lymphocyte ratio; LMR: lymphocyte-to-monocyte ratio; PLR: platelet-to-lymphocyte ratio; MEN-1: multiple endocrine neoplasia type 1; SSTR2: somatostatin receptor 2; MGMT: oxygen 6-methylguanine-DNA methyltransferase.

**Table 2 cancers-16-02502-t002:** Treatment strategies of 32 patients with Th-NETs.

Treatment		Number of Patients	%
Therapeutic regimens	Surgery	10	31.3
	Surgery + chemotherapy	7	21.9
	Surgery + radiation	1	3.1
	Surgery + chemotherapy +radiation	3	9.4
	Chemotherapy	7	21.9
	Chemotherapy + radiation	2	6.3
	Somatostatin analogs	1	3.1
	Everolimus	1	3.1
Surgery approach	Yes	21	65.6
	No	11	34.4
Adjuvant therapies (21 patients)	Chemotherapy	6	28.6
	Chemotherapy + radiation	3	14.3
	Radiation	1	4.8
	No adjuvant therapies	11	52.4
TMZ-based Chemotherapy (19 patients)	First-line treatment	5	26.3
	Second-line treatment	8	42.1
	Third or above line treatment	6	31.6

TMZ: temozolomide.

**Table 3 cancers-16-02502-t003:** Response to TMZ-based chemotherapy of 19 patients who received TMZ-based chemotherapy.

ID	Treatment Lines	MGMT Status	Ki67 (%)	Surgery	PFS (m)	OS (m)	Best Response
2	Second-line	-	10	Yes	22.0	38.2	Partial response
4	First-line	-	5	No	12.8	32.1	Partial response
5	Third or above line	-	10	No	13.7	35.6	Stable disease
6	Second-line	+	10	No	5.1	62.4	Stable disease
7	First-line	+	18	No	5.2	11.9	Stable disease
9	Second-line	+	25	No	7.0	11.1	Stable disease
13	Second-line	+	10	No	23.8	51.1	Stable disease
14	Third or above line	+	8	Yes	8.1	72.9	Stable disease
15	Second-line	+	8	Yes	23.9	60.9	Stable disease
16	First-line	+	20	No	12.5	41.3	Partial response
17	Second-line	+	5	Yes	7.9	29.6	Stable disease
19	Third or above line	+	15	Yes	13.2	94.3	Stable disease
21	Third or above line	+	40	Yes	2.7	30.1	Progressive disease
22	Second-line	+	10	No	6.0	19.3	Stable disease
23	First-line	+	40	No	4.1	34.2	Stable disease
28	Second-line	+	20	Yes	32.1	32.1	Stable disease
30	First-line	+	5	No	25.1	35.4	Stable disease
31	Third or above line	+	15	Yes	4.2	8.5	Progressive disease
32	Third or above line	+	10	Yes	5.7	5.7	Stable disease

**Table 4 cancers-16-02502-t004:** Univariable and multivariable analysis for overall survival.

Prognostic Factors		Univariable Analysis		Multivariable Analysis	
		HR (95% CI)	*p*	HR (95% CI)	*p*
Ki67	≤10%	1	0.004	1	0.007
	>10%	8.03 (1.56–41.45)		9.54 (1.55–58.58)	
NSE (ng/mL)	Low (<16.3)	1	0.025	1	0.448
	High (≥16.3)	4.96 (1.06–23.23)		2.28 (0.37–13.86)	
Distant metastasis	No	1	0.003	1	0.508
	Yes	6.93 (1.60–30.06)		2.39 (0.24–23.50)	
TNM stage	I–II	1	0.044	1	0.779
	III–IV	3.720 (0.96–14.45)		0.66 (0.06–7.03)	
Masaoka stage	I–II	1	0.044	1	0.779
	III–IV	3.720 (0.96–14.45)		0.66 (0.06–7.03)	
Surgery approach	No	1	0.002	1	0.003
	Yes	0.17 (0.05–0.59)		0.24 (0.02–3.92)	

NSE: neuron-specific enolase. Only significant variables (*p* < 0.05) upon univariable analysis are listed.

**Table 5 cancers-16-02502-t005:** Univariable and multivariable analysis for disease-free survival.

Prognostic Factors		Univariable Analysis		Multivariable Analysis	
		HR (95% CI)	*p*	HR (95% CI)	*p*
TNM stage	I–II	1	0.049	1	0.032
	III–IV	2.89 (0.96–8.70)		4.62 (1.14–18.66)	
Masaoka stage	I–II	1	0.049	1	0.032
	III–IV	2.89 (0.96–8.70)		4.62 (1.14–18.66)	
Adjuvant therapies	No	1	0.027	1	0.010
	Yes	0.28 (0.08–0.93)		0.13 (0.03–0.62)	
NLR ^a^	Low (<1.94)	1	0.020	1	0.046
	High (≥1.94)	3.72 (1.14–12.12)		3.59 (1.03–12.54)	

^a^ ROC curve analysis (AUC = 0.83) found that the optimal NLR cut-off value for the DFS was 1.94. NLR: neutrophil-to-lymphocyte ratio. Only significant variables (*p* < 0.05) upon univariable analysis are listed.

**Table 6 cancers-16-02502-t006:** Univariable and multivariable analysis for progression-free survival of patients treated with temozolomide.

Prognostic Factors		Univariable Analysis		Multivariable Analysis	
		HR (95% CI)	*p*	HR (95% CI)	*p*
Gender	Male	1	0.028	1	0.03
	Female	3.79 (1.06–13.6)		3.48 (0.88–13.80)	
NLR ^a^	Low (<1.94)	1	0.023	1	0.025
	High (≥1.94)	3.88 (1.12–13.42)		3.66 (0.97–13.81)	
PLR ^b^	Low (<106.09)	1	0.044	1	0.370
	High (≥106.09)	3.19 (0.98–)		1.82 (0.49–6.85)	

^a^ ROC curve analysis (AUC = 0.75) found that the optimal NLR cut-off value for the PFS was 1.94. ^b^ ROC curve analysis (AUC = 0.81) found that the optimal PLR cut-off value for the PFS was 106.09. NLR: neutrophil-to-lymphocyte ratio; PLR: platelet-to-lymphocyte ratio. Only significant variables (*p* < 0.05) upon univariable analysis are listed.

## Data Availability

The data presented in this study are available in the article and Appendix A.

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
