# Peer review of "Treatment, Prognostic Markers, and Survival in Thymic Neuroendocrine Tumors, with Special Reference to Temozolomide-Based Chemotherapy"

_cancers, 2024, doi:10.3390/cancers16142502_

Round 1

Reviewer 1 Report

Comments and Suggestions for Authors

In this paper, authors retrospectively investigated clinicopathological characteristics and therapeutic strategies in a monocentric series of thymic NETs. Due to the rarity of the disease, this paper provides useful data to discuss. Nevertheless, there are some points that need to be addressed.

It is stated that “biopsy and resection specimens were centrally reviewed by two expert 82 pathologists in neuroendocrine tumor pathology”, but do you have a multidisciplinary dedicated team to discuss also therapeutic strategies?

Considering the rarity of disease, were all patients screened for MEN1 or were they screened according to specific criteria? Please detail.

In results (page 3, line 145) please explain how the cut off of 7 cm has been chosen and detail tumor size in table 1.

Regarding patients treated with temozolomide (TMZ) In page 5, line 167-171 there is a description of several chemotherapy regimens done by these patients, but these data should be stratified according to surgery. In particular, table 3 should be modified, considering adding data about surgery with an additional column. Moreover, in the discussion (page 12 line339) it is not clear if TMZ therapy is a promising tool irrespective of sequencing, please detail. Furthermore, TMZ has been recently proven as effective second-line treatment in advanced neuroendocrine carcinomas (TENEC-TRIAL) and I suggest discussing these data (A new schedule of one week on/one week off temozolomide as second-line treatment of advanced neuroendocrine carcinomas (TENEC-TRIAL): a multicenter, open-label, single-arm, phase II trial. ESMO Open. 2024 May;9(5):103003. doi: 10.1016/j.esmoop.2024.103003. PMID: 38615472). Similarly, I suggest to expand the section about the clinical usefulness of NLR and other inflammatory markers discussing their role in other tumors and in particular in another rare neuroendocrine tumor, medullary thyroid cancer. 

Table 1: “Physical examination funding” please explain. Please detail the following “Carcinoids/NETs with elevated mitotic counts and/or Ki67 proliferation index” according to WHO classification. Furthermore, provide explanations about cut-off values of NSE, Pro GRP, NLR.

Table 2: in “Targeted therapy” author included both SSA and everolimus, while they should be separated.

Table S2: It should be pointed out that some drugs as surufatinib and tislelizumab are not widely available.

Acronyms should be revised, detailed only at their first use and then used consistently throughout the text (page 3, please detail ALC, NLR, PLR and LMR; revise TC, AC). References of RECIST and AJCC criteria should be added (page 2, line 94: RECIST criteria; page 3, line 105: AJCC).

Comments on the Quality of English Language

Minor language revision is required.

Author Response

Comments 1: It is stated that “biopsy and resection specimens were centrally reviewed by two expert 82 pathologists in neuroendocrine tumor pathology”, but do you have a multidisciplinary dedicated team to discuss also therapeutic strategies?

Response 1: Thank you very much for your kindness comments. I agree with this comment. I am pleased to share that our center established a dedicated neuroendocrine neoplasm clinic in 2010, and by 2012, we had assembled a multidisciplinary team. This team comprises physicians, surgeons, pathologists, radiation oncologists, and radiologists, each with over a decade of specialized experience. It is precisely the experts from this team who are involved in the management of treatment for patients with thymic neuroendocrine tumors. Therefore, I have added this part of the content on lines 92 to 97 on page 2 and marked it in red.    

Comments 2: Considering the rarity of disease, were all patients screened for MEN1 or were they screened according to specific criteria? Please detail.

Response 2: Thank you for pointing this out. I agree with this comment. Due to the uncommon occurrence of combined MEN1 syndrome and thymic neuroendocrine tumors, coupled with the significant expenses associated with genetic testing, we did not conduct genetic testing on all patients diagnosed with Th-NETs. But all patients with Th-NETs in this study underwent systematic evaluations and received clinical diagnoses in accordance with the Clinical Practice Guidelines for Multiple Endocrine Neoplasia Type 1 (MEN1). For patients presenting with a family history or two or more MEN1-associated endocrine tumors, such as parathyroid adenomas, gastroenteropancreatic neuroendocrine tumors, and pituitary tumors, genetic testing using the next-generation sequencing (NGS) method on blood samples was conducted. The detection of a MEN1 germline mutation confirmed the diagnosis of MEN1 syndrome. I have detailed the diagnostic approach to MEN1 syndrome on page 3 lines 132-139, and marked it in red. Meanwhile, in Table S1 of the supplementary material, I changed "MEN1" to "MEN1 syndrome" and used "Yes/No" to indicate whether the patient was diagnosed with MEN1 syndrome. This was done to avoid the ambiguity of confusing "MEN1 gene mutation" with "clinical diagnosis of MEN1 syndrome."

Comments 3: In results (page 3, line 145) please explain how the cut off of 7 cm has been chosen and detail tumor size in table 1.

Response 3: Thank you for the constructive comments, which greatly helped us to improve the manuscript. I have supplemented the average tumor size in Table 1 and detailed the tumor size for each patient in Table S1.

The tumor size cutoff of 7 cm previously reported in the Results (page 3, line 145) was chosen based on the experience from several large retrospective studies (1. Ose, N.; et al. Interactive CardioVascular and Thoracic Surgery 2018, 26, 18–24, doi:10.1093/icvts/ivx265. 2. Sullivan, J.L.; et al. Ann Thorac Surg 2017, 103, 935–939, doi:10.1016/j.athoracsur.2016.07.050. 3. Mao, Yinggang.; et al. Analysis of survival and prognostic factors of neuroendocrine tumors of the thymus. Journal of Basic and Clinical Oncology 2023, 36, 303–307.). However, based on your suggestions and upon revisiting the references, although we continue to use 7 cm as the cut-off value to analyze prognosis, we believe it is more appropriate to include this explanation in the discussion section. Therefore, we removed the statistical analysis section based on tumor sizes <7 cm and ≥7 cm from Table 1 and instead reported the average tumor size (page 4, lines 164-165). In the discussion section (page 11, lines 345-348), we added the rationale for selecting 7 cm as the tumor size cut-off value and discussed the implications of the findings.

Comments 4: Regarding patients treated with temozolomide (TMZ) In page 5, line 167-171 there is a description of several chemotherapy regimens done by these patients, but these data should be stratified according to surgery. In particular, table 3 should be modified, considering adding data about surgery with an additional column.

Response 4: Thank you for the constructive comments. Agree. I have removed the description of the chemotherapy regimen previously detailed on page 5, lines 167-171, and have reorganized the data to include stratification by surgical intervention. The newly stratified data is now presented on page 5, lines 186-192 and 194-199. I have updated Table 3 to include a new column detailing surgical interventions, with a corresponding description on page 5, lines 203-204Additionally, I analyzed the relationship between OS of patients who underwent TMZ-based chemotherapy and their surgical status (page 10, line 276). Unfortunately, the results from the multivariable analysis did not indicate that surgery is an independent prognostic factor.

Comments 5: Moreover, in the discussion (page 12 line339) it is not clear if TMZ therapy is a promising tool irrespective of sequencing, please detail.

Response 5: Thank you for pointing this out. I agree with this comment. Therefore, I have enhanced the discussion of the progression-free survival (PFS) of everolimus and somatostatin analogs (SSAs) in the treatment of thymic neuroendocrine tumors (Th-NETs) as a basis for comparison. This detailed comparison underscores that TMZ therapy is a promising option for managing these tumors (page 12, lines 396-405).

Comments 6: Furthermore, TMZ has been recently proven as effective second-line treatment in advanced neuroendocrine carcinomas (TENEC-TRIAL) and I suggest discussing these data (A new schedule of one week on/one week off temozolomide as second-line treatment of advanced neuroendocrine carcinomas (TENEC-TRIAL): a multicenter, open-label, single-arm, phase II trial. ESMO Open. 2024 May;9(5):103003. doi: 10.1016/j.esmoop.2024.103003. PMID: 38615472).

Response 6: Thank you for your recommendation. I have incorporated a detailed discussion of this clinical study into the Discussion section (page 12, lines 384-388) of the manuscript, which has significantly enhanced the substantiation of my argument.

Comments 7: Similarly, I suggest to expand the section about the clinical usefulness of NLR and other inflammatory markers discussing their role in other tumors and in particular in another rare neuroendocrine tumor, medullary thyroid cancer.

Response 7: Thank you for the constructive comments. I appreciate your insights, which have significantly enhanced the discussion section of my manuscript. I have expanded the section about the clinical usefulness of NLR in medullary thyroid cancer (page 12, lines 368-377). Because NLR is an inflammatory marker that demonstrated clear prognostic significance in our results. Unfortunately, due to space constraints, I was unable to provide a more detailed discussion on the application of other inflammatory markers in various tumors. However, I value your comments and plan to address this aspect in future research to further enhance the understanding of this topic. I appreciate your understanding and look forward to incorporating these elements in subsequent work.

Comments 8: Table 1: “Physical examination funding” please explain. Please detail the following “Carcinoids/NETs with elevated mitotic counts and/or Ki67 proliferation index” according to WHO classification. Furthermore, provide explanations about cut-off values of NSE, Pro GRP, NLR.

Response 8: Thank you for pointing this out. “Physical examination funding” refers to asymptomatic patients whose lesions were discovered through physical examinations. My previous statement was not accurate, so I changed “Physical examination funding” in Table 1 to “None a” and added a supplementary explanation below Table 1 as follows: “a Patients without symptoms had lesions detected during routine physical examinations. (page 5, line 172)”

I have detailed the following “Carcinoids/NETs with elevated mitotic counts and/or Ki67 proliferation index” in the materials and methods (page 3, lines116-120). And I have included explanations for the selection of NSE and Pro-GRP cut-off values in the notes under Table 1 (page 5, lines 173-177), and also added explanations for the selection of NLR and PLR cut-off values in the notes under Tables 5 (page 8, line 264) and Table 6 (page 10, lines 281-282).

Comments 9: Table 2: in “Targeted therapy” author included both SSA and everolimus, while they should be separated.             

Response 9: I agree with the comment. I have separated them in Table 2.

Comments 10: Table S2: It should be pointed out that some drugs as surufatinib and tislelizumab are not widely available.

Response 10: Thank you for pointing this out. Surufatinib’s clinical research was mainly conducted in China, so it is only recommended in Chinese guidelines for the treatment of advanced Th-NETs. And tislelizumab is not currently recommended in clinical guidelines for the treatment of Th-NETs. However, given the rapid progression observed in the patient's tumor and genetic test results indicating potential responsiveness to immunotherapy, we initiated treatment with Tislelizumab following informed consent from the patient's family. This approach aligns with a personalized treatment strategy based on specific genetic insights. So I have supplemented these two points below Table S2 (supplementary materials).

Comments 11: Acronyms should be revised, detailed only at their first use and then used consistently throughout the text (page 3, please detail ALC, NLR, PLR and LMR; revise TC, AC). References of RECIST and AJCC criteria should be added (page 2, line 94: RECIST criteria; page 3, line 105: AJCC).

Response 11: Agree. I have detailed ALC, NLR, PLR and LMR in materials and methods (page 3, lines 125-130), and revised TC, AC. I have added the references of RECIST and AJCC criteria (page 3, line 106, line 121).

Reviewer 2 Report

Comments and Suggestions for Authors

In this paper clinicopathologic characteristics and therapeutic strategies of Thymic neuroendocrine tumors are studied.

It is an interesting paper. I make the following observations:

I suggest expanding the introduction with more information on Th-NETs including updated references, such as that Th-NETs are part of thymic epithelial tumors, including the WHO classification.

In results, in my opinion it is better to highlight the relevant data in the text and not repeat the results that are already in the tables.

In the title of the continuation of supplementary table 1, the S is missing

I recommend that all the meanings of the abbreviations be noted at the bottom of the tables, so that they are not lost in the text.

In section 3.3 Survival, I recommend that the text write down the corresponding letter in figure 1 of what is being described.

The paragraph on lines 229 to 233 seems out of place

  It seems that reference 11 does not correspond to the text.

Author Response

Comments 1: I suggest expanding the introduction with more information on Th-NETs including updated references, such as that Th-NETs are part of thymic epithelial tumors, including the WHO classification.

Response 1: Thank you for pointing this out. I agree with this comment. Therefore, I have expended the introduction of Th-NETs (page 2, lines 49-50, lines 54-61) according to your comments and redlined.

Comments 2: In results, in my opinion it is better to highlight the relevant data in the text and not repeat the results that are already in the tables.

Response 2: Agree. I have, accordingly, removed some parts of the results that are not important to discuss, such as “The average ALC values were 1.6 ± 0.5 (range, 0.5-2.5), average NLR values were 2.2 ± 1.1 (range, 0.9-5.8), average LMR values were 4.1 ± 1.8 (range, 1.7-9.8), and average PLR value was 137.0 ± 56.3 (range, 30.9-295.9). (page 3, lines 139-142)” and “Additionally, fever was observed in 2 (6.3%) patients, while gastrointestinal symptoms or ectopic ACTH syndrome were each present in 1 (3.1%) patient. (page 3, lines 134-135)”

Comments 3: In the title of the continuation of supplementary table 1, the S is missing

Response 3: Thank you for pointing this out. I have added the “S” to the title of supplementary table 1 and redlined the modifications.

Comments 4: I recommend that all the meanings of the abbreviations be noted at the bottom of the tables, so that they are not lost in the text.

Response 4: Thank you for the constructive comments, which greatly helped us to improve the manuscript. I have noted all the meanings of the abbreviations at the bottom of the tables and redlined them (page 6, line 210; page 8, line 244; page 9, line 265; page 10, line 283; and some in Table S1 and Table S2 of the supplementary materials).

Comments 5: In section 3.3 Survival, I recommend that the text write down the corresponding letter in figure 1 of what is being described.

Response 5: Thank you for pointing this out. I agree with the comment. Therefore, I have changed the corresponding expressions in the original text to the corresponding letter in figure 1 of what is being described and redlined them (page 7, line 218, line 220, line 222, line 223).

Comments 6: The paragraph on lines 229 to 233 seems out of place

Response 6: Thank you for the constructive comments. Agree. This paragraph should really be separated from the discussion of patients receiving TMZ-based chemotherapy below, so I have moved it above Table 5 (page 8, lines 257-261).

Comments 7:  It seems that reference 11 does not correspond to the text.

Response 7: Thank you for pointing this out. I have revised the reference 16 to relevant literature that supports the points of the text, and added reference 17 to further support our points. The corresponding modifications are reflected in lines 506 to 511 on page 14 and are highlighted in red.
